# Real-Time Assessment of Mandarin Crop Water Stress Index

**DOI:** 10.3390/s22114018

**Published:** 2022-05-26

**Authors:** Sadick Amoakohene Appiah, Jiuhao Li, Yubin Lan, Ransford Opoku Darko, Kelvin Edom Alordzinu, Alaa Al Aasmi, Evans Asenso, Fuseini Issaka, Ebenezer Acheampong Afful, Hao Wang, Songyang Qiao

**Affiliations:** 1College of Water Conservancy and Civil Engineering, South China Agricultural University, No. 483, Wushan Road, Tianhe District, Guangzhou 510642, China; 20191000028@stu.scau.edu.cn (S.A.A.); kelvinedomalordzinu@stu.scau.edu.cn (K.E.A.); alaaasmi83@gmail.com (A.A.A.); whao20000904@gmail.com (H.W.); qsy1999@stu.scau.edu.cn (S.Q.); 2College of Engineering, National Center for International Collaboration Research on Precision Agricultural Aviation Pesticides Spraying Technology (NPAAC), South China Agricultural University, No. 483, Wushan Road, Tianhe District, Guangzhou 510642, China; ylan@scau.edu.cn; 3Department of Agricultural Engineering, University of Cape Coast, Cape Coast PMB, Ghana; ransford.darko@ucc.edu.gh; 4Department of Agricultural Engineering, University of Ghana, Accra P.O. Box LG 77, Ghana; easenso@ug.edu.gh; 5Soil, Water and Environmental Engineering Division, Soil Research Institute of Ghana, Kumasi PMB, Ghana; fuseini-issaka@csir.org.gh; 6Soil Science Division, Cocoa Research Institute of Ghana (Ghana COCOBOD), New Tafo-Akim P.O. Box 8, Ghana; ebenezer.afful@crig.org.gh

**Keywords:** plant-based indicators, real time, infrared thermometry, crop water stress indicator, Workswell Wiris Agro R infrared camera

## Abstract

The use of plant-based indicators and other conventional means to detect the level of water stress in crops may be challenging, due to their difficulties in automation, their arduousness, and their time-consuming nature. Non-contact and non-destructive sensing methods can be used to detect the level of water stress in plants continuously and to provide automatic sensing and controls. This research aimed at determining the viability, efficiency, and swiftness in employing the commercial Workswell WIRIS Agro R infrared camera (WWARIC) in monitoring water stress and scheduling appropriate irrigation regimes in mandarin plants. The experiment used a four-by-three randomized complete block design with 80–100% FC water treatment as full field capacity and three deficit irrigation treatments at 70–75% FC, 60–65% FC, and 50–55% FC. Air temperature, canopy temperature, and vapor pressure deficits were measured and employed to deduce the empirical crop water stress index, using the Idso approach (CWSI_(Idso)_) as well as baseline equations to calculate non-water stress and water stressed conditions. The relative leaf water content (RLWC) of mandarin plants was also determined for the growing season. From the experiment, CWSI_(Idso)_ and CWSI were estimated using the Workswell Wiris Agro R infrared camera (CWSI_W_) and showed a high correlation (R^2^ = 0.75 at *p* < 0.05) in assessing the extent of water stress in mandarin plants. The results also showed that at an altitude of 12 m above the mandarin canopy, the WWARIC was able to identify water stress using three modes (empirical, differential, and theoretical). The WWARIC’s color map feature, presented in real time, makes the camera a suitable device, as there is no need for complex computations or expert advice before determining the extent of the stress the crops are subjected to. The results prove that this novel use of the WWARIC demonstrated sufficient precision, swiftness, and intelligibility in the real-time detection of the mandarin water stress index and, accordingly, assisted in scheduling irrigation.

## 1. Introduction

Detecting crop water stress on time and precisely is very vital for both small- and large-scale precision agricultural systems, in order to avoid drought-related problems such as poor plant growth and development, reduction of yield, and yield quality [1,2,3]. In recent times, water scarcity issues are becoming widespread on a global scale, due to global warming resulting from the frequent emission of greenhouse gases, deforestation, and other environmental degradation factors, leading to erratic changes in climatic conditions. Rapid urbanization, massive industrial development, high population growth, and the increasing demand for fresh water for both domestic and commercial purposes are all significant threats to irrigated agriculture [4,5]. If unchecked, this scenario will bring about the production of crops under limiting water (water-stressed) conditions, with the resulting output being detrimental [2,6,7].

The most frequently used indicators for measuring water stress in tree crops are water potentials (Ψ) and stomatal conductance (gs), but these indicators are time consuming, very arduous, cannot be easily automated, and can only offer a point measurement in a leaf, a single soil location, a branch, or a tree [8,9,10]. Their measurements may not be true representations of the actual water status of an entire orchard field without a massive financial commitment, because several monitoring sites will be required in very large orchard setups. Furthermore, the information obtained from water potential (Ψ) measurements may not be consistent, because water potential (Ψ) decreases in plant species with isohydric characteristics (for example, in mandarin plants), due to their ability to preserve an effective stomatal regulation. This phenomenon impedes the noticeable decrease in water potential (Ψ) during limiting water (drought) conditions or during periods of high evaporative demand [10,11]. Change into right sign

Infrared thermometry has been employed from space to study the variation in plant canopy temperature as the condition of plant water varies [9,12,13]. However, the unavailability of geographic and time-based resolution of satellite images has limited the wider acceptance of the technology [14,15]. The research in [15] and [16] proved that equipping unmanned aerial vehicles (UAVs) with multispectral (IR and near-IR) cameras and correlating their data with data from the crop water stress index (CWSI) to study water status in olive and mandarin orchards was very effective. In commercial orchards, less difficult and automatic remote sensing technologies are required to rapidly and accurately estimate the water status of the plants, due to the large farm sizes and the estimated yield reduction resulting from water deficit [9,17,18,19]. It is worth noting that remote sensing technologies provide better observations in large established farms and easier means for managing the exact levels of plant water stress, thereby assisting subsequent operational decisions [20,21]. Several research projects have used less expensive and non-complex means of measuring canopy temperature (from infrared thermometers) and other related indicators such as crop water stress index (CWSI), the water potential index (Ψ), and the stomatal conductance index (I_g_) to monitor and study crop water level for several years [9,22,23].

The crop water stress index (CWSI) is a standardized index used to measure crop water stress and to minimize the possible impacts of environmental conditions affecting the interactions between plant canopy temperature and air temperature [12,24,25,26]. The measurement of the CWSI is established by the relationship between the actual canopy temperature (T_c_) data and the theoretical canopy temperature data, obtained from two different thresholds, namely a lower limit (when plant transpiration is maximum under full irrigation when stomata are completely opened) and an upper limit (when plants are not transpiring because they have been subjected to conditions of extreme water stress leading to complete closure of stomata) [9,26,27]. Furthermore, mandarin plant’s water stress can be determined by estimating the canopy temperature [28,29,30,31] at any given time. Nonetheless, the sensitivity of mandarins to air vapor pressure is very high, thereby minimizing the rate of transpiration under high VPD conditions and making the use of only canopy temperature very challenging [9].

Citrus water requirement is difficult to predict, due to differences in water consumption at different stages of growth, seasonal change, soil types, ground covers, ages and sizes of tree, and locations of cultivated plants. Different authors have described the different volumes of water needed by citrus. Mature citrus use about 100 to 511 L of water daily in summer [32,33,34,35]. In recent times, the CWSI model has been used to identify the water status in most crops. However, the procedures employed during the calculations of CWSI values in assessing plant water stress are very arduous and tedious, and involve a good deal of technical know-how, resulting in low farmer-researcher compliance [9,27,36,37].

To minimize the problems of farmer-researcher compliance and related issues, the Workswell Wirirs Agro R infrared camera has been designed to effectively evaluate crop water stress promptly, independent of secondary weather data, complex mathematical and scientific formulae, and workable operating system software. The WWARIC is designed to monitor crop water stress status in diverse farm situations, irrespective of the prevailing environmental conditions or the type of plants cultivated. This tool is very useful in on-farm water deficit estimation, irrespective of plant type, plants’ growth stages, and farm conditions. The aim of this experiment is to study the effectiveness and feasibility of using the WWARIC to monitor water stress and to assist in the appropriate scheduling of irrigation regimes for mandarin plants.

## 2. Materials and Methods

### 2.1. Experimental Site

The field experiment was conducted on a 576 m^2^ (0.142 acres) mandarin orchard that was already established at the South China Agricultural University Citrus Experimental Orchard near Yangcun town, Boluo county, Huizhou city of the Guangdong Province in the Peoples’ Republic of China, which has geographical coordinates of 36°16′2″ North and 115°28′32″ East, a mean annual rainfall of 1955 mm, and a mean annual temperature of 31 °C. The area is typical of a subtropical monsoon area, with the Tropic of Cancer crossing the city. The climate is mild and humid, with frequent rains from April to August [38].

The orchard trees were planted in 2014 at a spacing of 2.5 m within a block and 3.6 m between blocks. A four-by-three randomized complete block design (RCBD) was used for this field experiment. Each plot consisted of three sub-blocks with three mandarin plants each were studied, constituting a plant population of 36 distributed over 12 experimental blocks (plots). Each treatment was replicated three times. To minimize the possible effects on the treatments (especially the purposefully stressed treatments) of rainfall and evaporation of water from the soil, the demarcated field was covered with black woven polyethene mulch to ascertain, during the study period, whether the loss of irrigation water was only through transpiration and deep percolation. Figure 1 shows the layout of the experimental plot.

### 2.2. Soil Data

The sampled soil collected and analyzed revealed that the type of soil at the study field was loamy sand made up of 72.1% sand, 19.4% silt, and 8.5% clay. Soil samples were collected and tested before treatments. The soil’s physical and chemical properties are shown in Table 1.

### 2.3. Water Stress Treatments

Irrigation scheduling was based on recharging the volume of water in the soil back to the previous soil moisture content at field capacity before usage by the plant for photosynthesis and transpiration. Four different irrigation treatments, i.e., full irrigation (80–100% FC) and regulated deficit irrigation schedules (70–75% FC, 60–65% FC and 50–55% FC) were applied throughout the study period.

Water was supplied by means of a gravity-driven drip system with an adjusted flow rate of 6 L/h. Prior to each irrigation scheduling, the soil gravimetric water content was measured and expressed in terms of volumetric water content (θ_v_). These data were compared with the soil water content (θ_v_) data obtained by measuring with an IMKO TRIME-TDR probe that was inserted horizontally along the walls of ditches dug closed to the plant root zone (about 0.5 m from the base of the plant) at different depth intervals. The amount of irrigation water applied was estimated from the equation used in [39,40]:(1)$$d_{n}=\left(\frac{\left(\mathrm{FC}-\mathrm{SMC}\right)}{100}\right)\times \mathrm{BD}\times D$$
where FC = the field capacity of the soil, BD = the bulk density of the soil, D = the rootzone depth, and SMC = the prevailing soil moisture content on a volume basis, d_n_ = amount of irrigation water applied.

### 2.4. Fertilizer Application, Pest and Disease Management

In May 2021, the soil nutrient was improved by adding 4.5 kg h^−1^ of N-P_2_O_5_-K_2_O fertilizer (20-10-10 of nitrogen, phosphorus, and potassium, respectively) as a top-dressing 30 cm from the plant stem into the soil, prior to treatment allocation. The farm was often sprayed with a mixture of Fenproximate, Azadirachtin, Bacillus thuringiensis, and other organic pesticides and insecticides to minimize the destruction caused by pests and insects, through a rotary center-pivot overhead mist sprayers mounted on metal pipes about 5 m high at 5 m intervals across the length and breadth of the field.

### 2.5. CWSI Workswell Wiris Agro R Infrared Camera

#### 2.5.1. Camera Specification, Data Collection, and Processing

The WWARIC is designed to be mounted on UAVs for assessing the crop water stress index (CWSI) as well as the temperature. The camera, endowed with a powerful operating system (WIRIS OS), ensures full accessibility and utilization of all the camera’s operational functions during instantaneous data streaming and assessment, for onboard operations or offboard use. The WWARIC can be operated simply through the MavLink protocol, the RJ-45 connector, the S. Bus system, the CAN bus system, or a triggers support system. With a “plant biomass coverage” index, this camera displays the percentage of a real-time calculated mass of vegetation in the RGB setup. The CWSI, estimated with the WWARIC, is normalized within the limit of 0 to 1 (or 0–100%, with 100% being very stressed conditions while movement towards 0 represents less stressed conditions), representing the extent of stress on the crop in relation to pixel value. This information is vital in developing yield maps and making precise water management decisions in large-scale agriculture environments.

The Workswell Wiris Agro R infrared camera has a long-wave infrared band sensor with a specific sensor resolution of 640 by 512 pixels and a focal-plane array (FPA) active sensor with a size of 1088 × 10^−5^ by 8705 × 10^−6^ m. The field of view of the WWARIC lens is 45° and the standard temperature responsiveness is 0.05 °C (50 mK). The camera comes with four color maps: crop pallets, cropstep pallets, water pallets, and waterstep pallets that are used to manage and assess the CWSI. The WWARIC can perform digital zooming of images up to 14 times continuously (i.e., 1–14×). The camera uses the advanced Workswell CorePlayer application software package for offline CWSI data analysis, which comes with two licenses. The 3D mapping software works perfectly with Agisoft and Pix4D softwares. Operationally, the 10× optical zoom RGB has a resolution of 1920 × 1080 pixels (full high-definition), a $\frac{1}{3}$ inch sensor, automatic white offset, a powerful varied range, backlight counterbalance, exposure, and gamma control systems. The WWARIC has 3.0 to 350 mm focal length and 10× optical zoom that supports vibration and a view angle as low as 6.9° (for ultra zoom) with a maximum of 58.2° (for extra-wide viewing). The WWARIC has an auto-focus function that synchronizes a direct focus zoom, an in-built high-speed solid-state drive with 128 GB capacity for storing captured images and videos, an exterior secure digital card slot, and a USB 2.0 port.

The WWARIC stores images and videos in the following layouts: JPEG images, TIFF images, digital camera full high-definition JPEG images, and digital camera h.264 encode video high-definition recordings and full-frame video recordings (raw data recordings). With an average mass of 4.30 kg and dimensions of 8.30 cm long, 8.50 cm wide, and 6.80 cm high, the camera can be fixed on a 2 × $\frac{1}{4}$ − 20 UNC support system connected to a tripod stand, an unmanned aerial vehicle, or a satellite system. The camera is encased in a robust aluminum plate that can endure external temperature fluctuations, and which serves as a protector for the internal parts against being damaged, to ensure long-term measurement stability. The operating and storage temperatures are, respectively, a minimum of −10 °C and a maximum of 50 °C and a minimum of 30 °C and a maximum of 60 °C. The camera has an input power source of 9–36 V DC, with a coaxial cable of 2 × 6.4 mm and an outer shell GND that dissipates 12 W of power.

#### 2.5.2. Camera Measurement Functions

The Workswell Wiris Agro R infrared camera has an inbuilt four-color map with full radiometric (temperature) information. The camera can be operated rapidly to capture videos and images simultaneously as thermal imagery, digital imagery, or digital video imagery. During operation, the camera images can be viewed as picture-in-picture (PIP), full screen RGB with segmentation, or dual-screen. There are three micro-HDMI video outputs in the formats: 1280 × 720 pixels (720p), an aspect ratio of 16:9, and a micro-HDMI video output.

### 2.6. Field Measurements

#### 2.6.1. Image Acquisition

The WWARIC, purchased from Albrechtova Vrchu, Prague, Czech Republic, was mounted on a metallic pole, 12 m high, to capture images of mandarins cultivated under different soil water contents at various field capacities to estimate the water levels in the mandarin plants. The CWI-640-545-128 camera specifications are as follows: it was released on 20 January 2021 with revision number 210120, serial number 1010-CWI-201002, version 1.3.8, and resolution 640 × 512. The RTC temperature is 37.6 °C, with CPU and IR core temperatures of 46.3 °C and 45.8 °C, respectively. Data gained by the WWARIC were processed with CorePlayer software version 1.4.1. The CWSI_W_ was compared with the CWSI_(Idso)_ and the relative leaf water content (RLWC) to determine precision through correlation.

#### 2.6.2. Canopy Temperature Measurement

The mandarin canopy temperature was calculated with the help of a Raytek^®^ portable, noncontact, infrared thermometer having an adjusted emissivity of 0.95 Wm^−2^ with a 0.2 °C (0.5 °F) display resolution. The measurement was done by holding the IR thermometer 0.30 m directly above the mandarin leaf, with the optical laser spot directed vertically to the targeted leaf [20,41]. The temperature readings were recorded once every 7 days after July 2021, when the first irrigation dose was administered until harvesting. The mandarin leaf temperatures of three sampled plants per treatment were recorded from four directions, with north as the initial point of reference, and the average temperatures of each treatment were estimated. The temperature recordings were usually measured on very clear sky sunny days between 11:00 A.M. and 14:00 P.M., on fully developed sunlit mandarin leaves at the tip of the branches, as proposed by [42,43,44].

#### 2.6.3. Estimating the Relative Leaf Water Content (RLWC)

The RLWC was estimated every week by weighing freshly plucked sampled mandarin leaves to deduce their fresh weight (FW), subsequently soaking and bottling them in distilled water at 25 °C for 144 h to ensure full turgidity, and then weighing them again to determine the weight when fully turgid (TW). The turgid leaves were cleaned, oven-dried at 80 °C for 48 h and reweighed to deduce the dried weight (DW). The relative leaf water content (RLWC) was then estimated according to the equation used by [20,45]:(2)$$\mathrm{RLWC}=\left(\frac{\left(\mathrm{FW}-\mathrm{DW}\right)}{\left(\mathrm{TW}-\mathrm{DW}\right)}\right)\times 100$$
where RLWC (%) = the relative leaf water content, FW = the weight of fresh leaves, DW = the weight of dried leaves, and TW = the weight of turgid leaves.

#### 2.6.4. Estimating SPAD and the Nitrogen Sufficiency Index (NSI)

The chlorophyll content of each leaf was measured with a portable Konica Minolta SPAD 502 chlorophyll meter. The SPAD measurements were made on new sampled and completely opened leaves every week, from the beginning of irrigation. The measured SPAD values were converted into the nitrogen sufficiency index (NSI) by finding the ratios of the measured leaf chlorophyll concentration in each plant for each treatment to the measured chlorophyll concentration of a non-stressed reference plant, on the same day of measurement as described by [46] in the following equation:(3)$$\mathrm{NSI}=\frac{{\mathrm{SPAD}}_{M}}{{\mathrm{SPAD}}_{R}}$$
where NSI is the nitrogen sufficiency index, SPAD_M_ is the measured chlorophyll content in the observed plant, and SPAD_R_ is the measured chlorophyll content in the fully irrigated reference mandarin crop.

#### 2.6.5. Estimating the CWSI Using an Empirical Method (the Idso Method)

The CWSI can be determined experimentally by studying the differences in plant canopy temperature and air temperature (T_c_ − T_a_), as well as considering the vapor pressure deficit (VPD) of the air on a well-irrigated plant on a clear, sunlit, unclouded day with maximum plant environmental settings [27,36,37]. In calculating the CWSI, two focal points were designated and used as a lower-referenced point (the non-water stressed baseline dependent on VPD) and an upper-referenced point (the non-transpiring canopy/water stressed baseline not dependent on VPD) [27,36,37,47,48,49,50].

The CWSI empirical value is estimated using the equation applied in [36]:(4)$${\mathrm{CWSI}}_{\left(\mathrm{Idso}\right)}=\frac{(T_{C}-T_{a})-D_{2}\;}{D_{1}-D_{2}}$$
where T_c_ is the mandarin leaf cover temperature (°C); T_a_ is the orchard air temperature (°C); D_2_ = the lower reference point ($(T_{C}-T_{a})\times \mathrm{VPD}$ for well-watered mandarin plants, i.e., non-stressed conditions obtained from 80–100 FC treatment); and D_1_ = the upper reference point (the difference in canopy and air temperature difference for highly watered mandarin plants (extremely water stressed). The upper reference point was determined from canopy temperature data obtained from fully stressed mandarin plants (50–55% FC treatment) during the peak of the summer period when the maximum transpiration was envisaged [47,48]. The weather data used to calculate the CWSI by Idso procedures are presented in Table 2.

This CWSI empirical method, or CWSI_(Idso)_, expresses the plant water level within the range of 0 to 1, indicating no or less water stress levels and high-water stress status, respectively. However, negative values may also be obtained, indicating the presence of excess water.

#### 2.6.6. Estimating the CWSI with the Workswell WIRIS Agro R Infrared Camera

The WWARIC is designed to estimate the crop water stress index via three different methods. The CWSI model differential (CWSI_d_) mode represents the fundamental method that employs the measurement of air temperature (T_air_) as the only parameter during estimation. The CWSI_d_ value is estimated from the differences in the crop canopy temperature and the prevailing air temperature. The second method, the CSWI model empirical (CWSI_e_) mode, is based on measuring the temperatures of two reference points, T_wet_ (representing no stress or a 0% stress level) and T_dry_ (representing fully stressed or a 100% stress level). The CWSI model theoretical (CWSI_t_) mode is the third model that estimates the CWSI using the following parameters: the category of plant, the canopy temperature (°C), the air temperature (°C), the relative humidity (%), the baseline slopes, and the baseline intercept.

### 2.7. Data Analysis

Statistical analyses were conducted using the IBM Statistical Package for Social Sciences version 21 software. One-way ANOVA was employed to analyze the differences in the water stress parameters studied (the leaf temperature (LT°C), the relative leaf water content (RLWC) (%), the leaf chlorophyll content (LCC) (mg g^−1^), and the water stress index, CWSI_(Idso)_) within treatments and blocks. The treatment means were separated by the Tukey HSD method to indicate significant differences among the treatment means, at 0.001, 0.01, and 0.05 levels of significance. Linear regression analysis was used to show the relation between soil volumetric water content and soil water content, measured with the time domain reflectometer, as well as the water stress indices obtained by both the Idso approach and the WWARIC.

## 3. Results

### 3.1. Soil Water Content Measurement

This study showed that the estimated volumetric soil water content and the TDR readings were strongly correlated. The correlation coefficient (R^2^) was found to be R^2^ = 0.93 at *p* < 0.05, as shown in Figure 2. The total irrigation water supplied to the mandarins from July 2021 to 30 November 2021, was 184.64 mm, 134.02 mm, 116.40 mm, and 100.14 mm at 80–100% FC, 70–75% FC, 60–65% FC, and 50–55% FC, respectively.

#### Variations in Soil Moisture Content

Fluctuations in weather conditions resulted in a corresponding variation in soil moisture content throughout the entire research period; hence, there were variations in the depths of the irrigation water supplied. The study showed that the irrigation water supplied was highest for all treatments during mid-summer (from 3 August 2021 to 31 August 2021) when transpiration was very high due to high daily temperatures. In contrast, the depth of the irrigation water supplied was low in July (when irrigation was started) for all the treatments but increased after July. In autumn, there was a decrease in the depth of the irrigation water supplied, resulting from the decrease in temperatures. The irrigation amount supplied and the weekly variation in irrigation water are presented in Table 3 and Figure 3, respectively.

### 3.2. Plant-Based Indicators

#### 3.2.1. Relative Leaf Water Content (RLWC)

The relative leaf water content increased with increasing volume of water available to the plant. The research showed that there was a gradual increase in RLWC across all the treatments from the beginning of irrigation treatment, notwithstanding that there were few weeks where there was little decline in the values observed. The estimated average relative leaf water content (%) during the study period was 93.93, 86.79, 84.59, and 82.88 for treatments 80–100% FC, 70–75% FC, 60–65% FC, and 50–55% FC, respectively. The observed relative leaf water content for the study period is presented in Figure 4.

#### 3.2.2. Leaf Temperature (°C)

The plant canopy temperature (leaf temperature) varied within treatments and between treatments. The average seasonal values recorded were 34.49, 35.64, 36.36, and 37.01 for the fully irrigated plants (80–100% FC) and the deficit irrigated treatments (70–75% FC, 60–65% FC, and 50–55% FC), respectively.

#### 3.2.3. Leaf Chlorophyll (mg g^−1^) Content and the Nitrogen Sufficiency Index

The weekly SPAD dataset measured during the period were highly significant. The observed SPAD values increased with the increasing soil water content at field capacity. The average SPAD values estimated were 72.3, 69.4, 68.5 and 65.4 mg g^−1^ for treatments 80–100% FC, 70–75% FC, 60–65% FC, and 50–55% FC, respectively. The average NSI values observed were 1, 0.948, 0.935, and 0.894 for treatments 80–100% FC, 70–75% FC, 60–65% FC, and 50–55% FC, respectively.

### 3.3. Mandarin Crop Water Stress Index (CWSI) Based on Idso Procedures and Baseline Equations

The CWSI values varied under the different irrigation treatments imposed on the plants from July 2021 to November 2021. The vapor deficit pressure was estimated from the relationship developed by [51] and utilized by [26,52], as follows;
(5)$$\mathrm{VPD}=0.6108\times \exp\;(\frac{17.27\;\left(T_{a}\right)}{\left(T_{a}+237.3\right)})\times \left(\frac{100-\mathrm{RH}}{100}\right)$$
where VPD = vapor deficit pressure T_a_ = the air temperature and RH = the relative humidity.

The CWSI empirical value was subsequently calculated from the relationship between VPD and the corresponding canopy and the air temperature difference (T_c_ − T_a_) was calculated from the relationship described previously in Equation (4).

Weekly disparities were observed in both summer and autumn with regard to VPD, as were differences in the canopy temperature and the air temperatures measured during the period. The average seasonal T_c_ − T_a_ (°C) values recorded in summer 2021 were 2.23, 2.41, 2.57, and 2.70 for treatments 80–100% FC, 70–75% FC, 60–65% FC, and 50–55% FC, respectively, with their corresponding average VPD (kPa), monitored for the same period, at 1.31, 1.47, 1.58, and 1.69, respectively. In autumn, the observed mean seasonal T_c_ − T_a_ (°C) values measured for the treatments 80–100% FC, 70–75% FC, 60–65% FC, and 50–55% FC were 2.21, 2.36, 2.50, and 2.63, respectively, with their corresponding VPD at 1.28, 1.44, 1.53, and 1.65, respectively. Figure 5A–D shows the weekly disparities in T_c_ − T_a_, as well as the VPD during the study period.

The baseline equations for the mandarins were established as $\mathrm{Tc}-\mathrm{Ta}=-0.56\times \left(\mathrm{VPD}\right)+4.05$ for the non-water stress baseline and Tc−Ta=4.05 for non-transpiring (i.e., extremely water stressed conditions) baseline equations. During the establishment of these baseline equations, all data collected were put together to get a fair representation of the weather data available. Correlating VPD and T_c_ − T_a_ datasets for the non-transpiring treatments gave rise to the lower baseline equation and VPD against T_c_ − T_a_ data values for the extremely stressed treatments resulted in the development of the upper baseline equation. T_c_ − T_a_ had a significant correlation with VPD at R^2^ = 0.58 for the non-stressed treatments while it was weakly correlated to VPD at R^2^ = 0.005 for the highly stressed treatments. These are shown in Figure 6A,B.

The computed seasonal CWSI_(Idso)_ values ranged from −0.21–0.79, with average seasonal values of 0.14, 0.33, 0.48, and 0.61 recorded for treatments 80–100% FC, 70–75% FC, 60–65% FC, and 50–55% FC, respectively, in summer, while the average seasonal values for autumn were 0.16, 0.32, 0.46, and 0.58 for the treatments 80–100% FC, 70–75% FC, 60–65% FC, and 50–55% FC, respectively. These values are shown in Figure 7A,B.

Both the VPD and the measured canopy-air temperature differences were significantly different (at *p* < 0.001) among the treatments and within treatments; however, both parameters were strongly correlated to the CWSI, as determined from the Idso procedures at R^2^ = 0.67 and R^2^ = 0.79, respectively, as shown in Figure 8A,B.

### 3.4. Mandarin CWSI Estimated from the Workswell WIRIS Agro R Infrared Camera

The results obtained from the WWARIC in estimating the mandarin CWSI followed a similar pattern and showed consistency with the results when the CWSI by Idso procedures were adopted to calculate the mandarin CWSI. Figure 9, Figure 10 and Figure 11 show a digital image of the mandarin orchard captured using the WWARIC, the isothermal graph, and the CWSI image, as well as a 3D graph gained from the WWARIC after analysis with the WIRIS core player version 1.4.1 during a data collection session in the autumn of 2021.

The outcome of the estimated CWSI from using the WWARIC on the four water application levels was established in autumn 2021 by categorizing the CWSI values into a 0.25 interval range, comparable to the procedures proposed by [24,53,54]. The CWSI recorded during the measurement session under the 80–100% FC treatment was within the CWSI range of 0.00 < CWSI_W_ *≤* 0.25, which is shown by a deep green color, while treatments under the 70–75% FC water stress level were within the range 0.25 < CWSI_W_ *≤* 0.5, represented with a light green color. In addition, the CWSI for mandarins under the 60–65% FC water stress level was within the 0.5 < CWSI_W_ *≤* 0.75 range, which is represented by a yellow color, while the CWSI for mandarins under the 50–55% FC water stress level was within the range of 0.75 < CWSI_W_ *≤* 1.00, which is represented with a light brown color. Nevertheless, the CWSI value > 1, indicated with the dark brown color on the TIFF image, shows the mulch cover, but not necessarily water stress within the canopy, as seen in Figure 10.

### 3.5. CWSI Based on the Idso Method and the WWARIC

The outcome of this research showed a high relationship between the CWSI estimate based on the Idso procedure, by relying on the field measurements of the mandarin canopy temperatures measured via the portable Raytek^®^ handheld, non-contact, infrared thermometer, and the CWSI estimated from using the WWARIC, during the study period with a correlation coefficient of R^2^ = 0.75, as seen in Figure 12.

### 3.6. The CWSI and Plant-Based Water Stress Indicators (RLWC, Leaf Temperature, and Chlorophyll Content)

#### 3.6.1. The CWSI Calculated by the Idso Method and Plant-Based Water Stress Indicators (RLWC, Leaf Temperature, and Chlorophyll Content)

The research revealed a positive connection between relative leaf water content, the chlorophyll content, and the soil water content, while an inverse relationship existed between the soil water level and the measured leaf temperatures. Generally, increasing the soil water content resulted in an increase in both the RWLC and the chlorophyll content, and vice versa. Mandarins subjected to no stress (80–100% FC) recorded the highest values for both the RWLC and the chlorophyll content, and vice versa. However, mandarin leaves subjected to this same treatment (i.e., 80–100% FC) recorded the lowest temperature measurements. Nonetheless, all three of the studied water stress indicators (RLWC, LT, and LCC) were significantly correlated with the CWSI estimated by the Idso approach at R^2^ = 0.68, 0.51, and 0.80, respectively, as seen in Figure 13A–C. Nevertheless, the calculated CWSI_(Idso)_ was inversely related to both the RWLC and the leaf chlorophyll content, but directly related to the leaf temperature. An increase in the CWSI_(Idso)_ corresponded to lower values of the RWLC and the measured leaf chlorophyll, and vice versa. On the other hand, the CWSI calculated by the Idso procedure increased with the increasing leaf temperatures recorded.

#### 3.6.2. The CWSI Estimated from the WWARIC and Plant-Based Water Stress Indicators (RLWC and Leaf Temperature)

The research showed that the CWSI estimated with the WWARIC was significantly correlated with the RLWC and the leaf temperature, with R^2^ values of 0.77 and 0.79, respectively. This correlation is shown in Figure 14A,B. The mandarins’ RLWC, and the estimated CWSI from the WWARIC on mandarins from the field, depicted a negative relationship. It was observed that at high values of mandarins’ RLWC, there were corresponding lower values of the CWSI estimated from the WWARIC, and vice versa. Furthermore, the recorded leaf temperatures varied directly with the values of the estimated CWSI gained with the WWARIC, leading to a positive significant relationship. It was observed that at low measured leaf temperatures, the corresponding CWSI values gained from the camera were also low, and vice versa.

## 4. Discussion

The crop water stress index has been useful in numerous agronomic researches to determine plant stress caused by water and/or soil-water deficit conditions [12,13,25,26,52,55]. This concept was first established and proposed by Idso and his research team, who adopted and worked with environmental factors, such as prevailing vapor deficit pressures and air temperatures [36]. The Idso method has been subsequently adopted, modified, and applied in other related studies to monitor plant-water stress related conditions. The results from these previous studies indicated a very strong relationship between the CWSI calculated using the Idso approach and the CWSI calculated using other procedures [25,41,56,57,58]. Nonetheless, the arduous nature of obtaining the required metrological data, the difficulty in computations, and the high complexity, advanced technicality, and time involved in estimating the crop water stress index by the Idso approach made it challenging at the farm level to assess crop water and soil-water conditions to efficiently control drought and yield loss; hence, the need arose for most agronomists to resort to using remotely sensed data obtained from very high-resolution cameras to resolve these problems [49,52,56]. These studies showed that the baseline equations established for mandarins ($\mathrm{Tc}-\mathrm{Ta}=-0.56\times \left(\mathrm{VPD}\right)+4.02$) and (Tc−Ta=4.02) for the non-water stress baseline and for the non-transpiring (extremely water stressed conditions) baseline equations, respectively, were similar to the equations developed by [59]. Nevertheless, although the linear regressions developed for the non-water stress baseline from the T_c_ − T_a_ and VPD datasets were significant at R^2^ = 0.58 (*p* < 0.05), the observed data were scattered. This outcome was consistent with the findings from [59,60,61], which observed that differences in the canopy-air temperatures and the VPD values measured for mandarin to develop the non-water stress baseline have a more scattering characteristic when compared to values of T_c_ − T_a_ and VPD recorded for developing non-water stress baseline equations for different orchard crops, such as sweet lime and pistachio. These authors attributed this phenomenon to variations in the mandarin canopy temperature.

The results further revealed that the CWSI computed using the Idso approach increased with an increasing soil-water deficit. The average estimated CWSI from the Idso approach for the period of study was in the order of 80–100% FC < 70–75% FC < 60–65% FC < 50–55% FC, which corresponded to CWSI values of 0.15 < 0.32 < 0.47 < 0.60. These figures are not different from the results of [27,52,54,61,62], where it was observed that the CWSI in plants increases with a decreasing availability of water in the soil (or increasing water stress). Furthermore, the CWSI calculated from the Idso approach was correlated with both the VPD and the difference in canopy-air temperature. This is consistent with the results from [62,63], where it was argued that the CWSI calculated by the Idso approach was a more sensitive index to the VPD and the canopy-air temperature difference, compared to other indices such as degree of non-stress and degrees above canopy threshold. Furthermore, the inverse relationship between estimated CWSI by the Idso approach and the RWLC from our studies is similar to the results from the works of [64,65,66], in their studies of soybean and corn. The leaf temperature was also directly related to the calculated CWSI, which is also consistent with studies from [59,62,67]. The effects of water stress on the leaf chlorophyll content were observed during the study period. This result showed a strong correlation between the calculated CWSI and the measured leaf chlorophyll content. This correlation is not different from the reports in [66,68,69,70,71], where it was observed that increasing crop water stress decreases the leaf chlorophyll content in a plant.

The strong association between the estimated CWSI from the field data and the CWSI gained by the WWARIC is an indication that the camera is capable of estimating the level of stress a particular plant is subjected to in real time and also in any plant setting. This finding is comparable to the studies in [24], where the WWARIC was used to successfully study water stress in tomato grown under greenhouse conditions. In estimating conventional CWSI by any method, a meteorological dataset is required before computations can be done. This requirement limits its use in modern-day precision agriculture, where the Internet of Things and big data systems, as well as artificial intelligence, make life easy. It is with this effect that the WWARIC has been endowed with all the features making it possible and easy to measure the correct prevailing air and canopy temperatures, humidity, and other parameters simultaneously to estimate water stress precisely in plants under any plant condition, from small fields to vast open fields and/or inside greenhouses in real time Other commercial infrared cameras and sensors do not have these features, providing the WWARIC with an advantage over them as far as in situ, instantaneous, and precise CWSI measurements are concerned. Furthermore, the works of [72,73,74,75,76] indicated that most thermal infrared cameras and sensors require surrogate calibrations before temperature data can be retrieved and utilized for subsequent determination of the CWSI. The WWARIC needs no calibration before it can measure the temperature and determine the CWSI of the plant under consideration. In addition, the in-built color map feature allows for easy identification of the temperature and the CWSI values of plants within specific regions of the field. It is worth noting that all three modes of the camera, differential, empirical and theoretical, were able to detect water stress in the mandarin orchard at heights above 12 m. The WWARIC is affected by the clouds, as the presence of a cloudy weather reduces the reflections on the plant canopy, thereby obscuring clear image acquisition; hence, it is advisable to use the camera on a clear sunny day, taking the measurements for calculating the CWSI with the Idso procedure.

## 5. Conclusions

The use of thermal indices based on canopy temperature for crop water stress surveillance and irrigation management has been established by numerous scientists in the last few decades, following the introduction of portable infrared thermometers. Since then, scientists have designed various scientific equipment and tools for standardizing the effects of some variable meteorological parameters, including VPD, wind speed, solar radiation, air temperature, etc. This novel research utilized a commercial WWARIC to remotely monitor mandarin CWSI precisely and instantaneously in real time, to swiftly model and schedule irrigation appropriately. Using the WWARIC, the predictable CWSI in mandarin development at each water deficit treatment level was shown by the following color codes: deep green representing the treatment 80–100% FC (little or no stress) within a range of 0<CWSI≤0.25; light green showing 75–70% FC (mild to medium water stress level) for the range 0.25<CWSI≤0.5; yellow representing 65–60% FC (medium to high water stress level) within the range 0.5<CWSI≤0.75; and light brown depicting 55–50% FC (high to extremely high water stress level) for the range 0.75<CWSI≤1. In contrast, dark brown colors, seen on the TIFF image, represent complete drought with no water present (CWSI>1), which in this case was the image of the ground plastic mulch cover, since there was not a total drought situation in the study field. In this research, all three pixel modes (empirical, theoretical, and differential) of the WWARIC were used to establish the CWSI at a height of 12 m. The CWSI_W_ values found in this study were consistent with the results obtained from the CWSI_I_ and with the relative leaf water content (RLWC), as well as with the leaf chlorophyll content of the mandarin plant. However, the WWARIC was designed to estimate CWSI (a standardized index established to compute water stress) while subduing the effects of environmental factors that affect the relationship between stresses, canopy temperature, and air temperature, unlike other infrared sensors and cameras. This makes this WWARIC a better option in determining the CWSI in plants, while minimizing the effects of several environmental factors. The CWSI obtained using the WWARIC is suitable for monitoring crop water conditions in real time to overcome drought and making immediate management decisions to reschedule irrigation earlier to avoid plant death and yield loss, while maximizing water use efficiency for mandarin production. This novel use of the WWARIC has demonstrated sufficient precision, swiftness, and intelligibility in the real-time measurement of mandarin water stress, depicting estimated CWSI in specific color ranges and values. When adopted and used in precision agriculture, this camera will save significant time, money, and other resources.

## Figures and Tables

**Figure 1 sensors-22-04018-f001:**
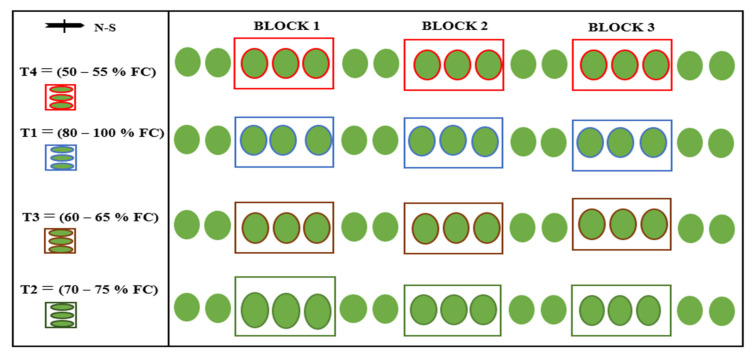
Field layout and experimental design.

**Figure 2 sensors-22-04018-f002:**
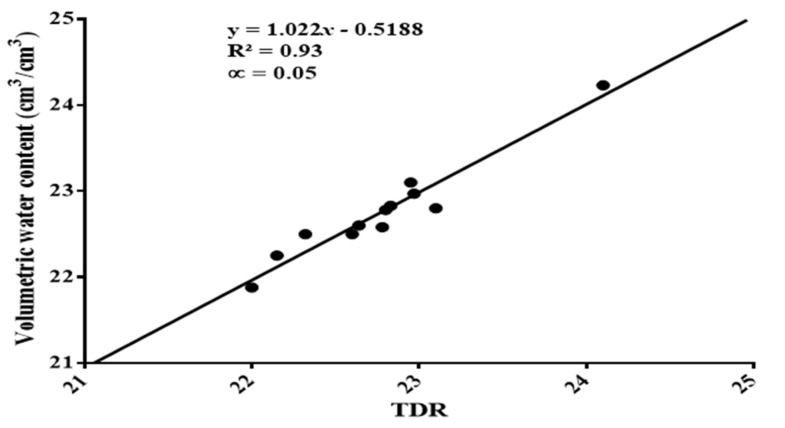
Correlation between soil volumetric water content and TDR readings.

**Figure 3 sensors-22-04018-f003:**
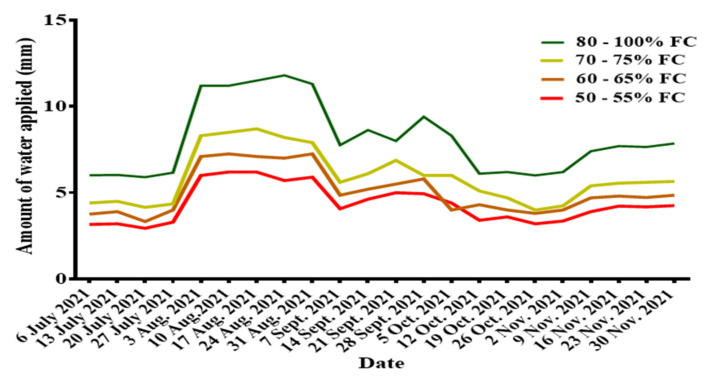
Variations in soil water content.

**Figure 4 sensors-22-04018-f004:**
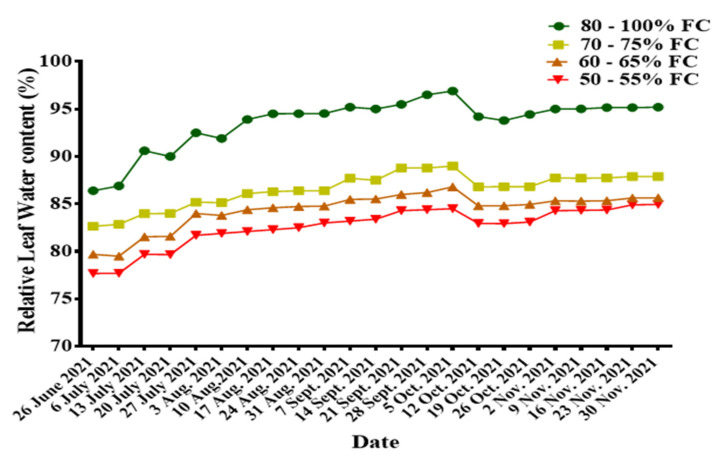
Seasonal variations in relative leaf water content.

**Figure 5 sensors-22-04018-f005:**
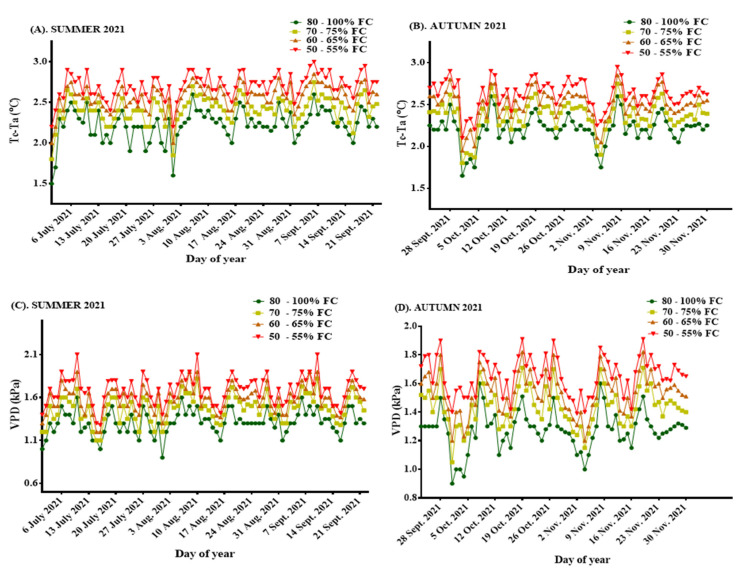
(**A**) Weekly T_c_ − T_a_ (°C) for summer, 2021; (**B**) Weekly T_c_ − T_a_ (°C) for summer, 2021; (**C**) Weekly VPD (kPa) for autumn, 2021; (**D**) Weekly VPD (kPa) for autumn, 2021.

**Figure 6 sensors-22-04018-f006:**
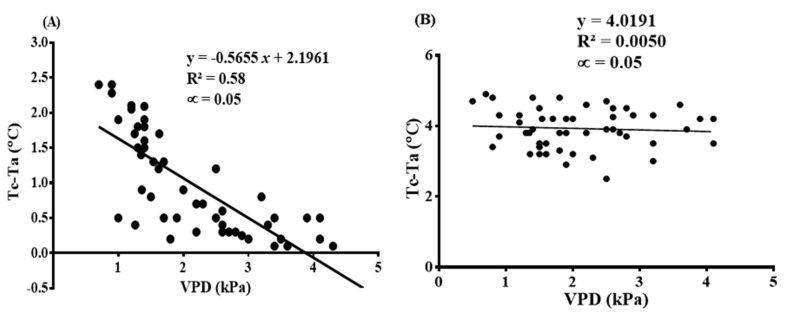
Relationship between T_c_ − T_a_ (°C) and VPD (kPa) for developing (**A**) non-water stressed and (**B**) non-transpiring baseline reference points in relation to fully irrigated and extremely stressed mandarin trees at noon on a very bright, clear day.

**Figure 7 sensors-22-04018-f007:**
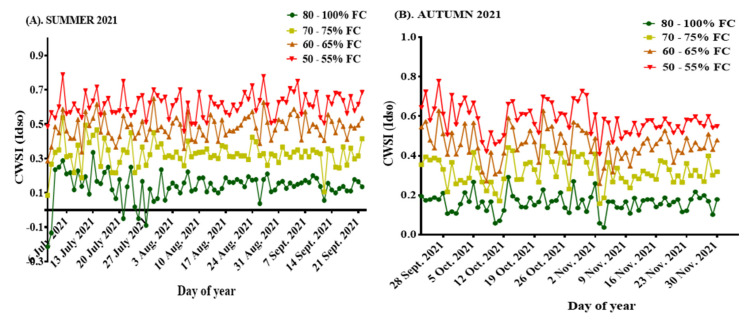
(**A**) CWSI for summer 2021; (**B**) CWSI for autumn 2021.

**Figure 8 sensors-22-04018-f008:**
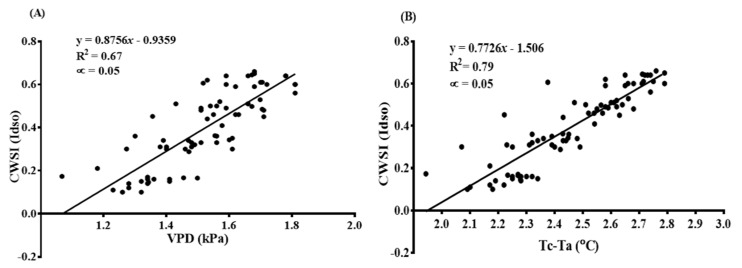
Correlation between (**A**) CWSI_(Idso)_ and VPD (kPa); (**B**) CWSI_(Idso)_ and T_c_ − T_a_ (°C).

**Figure 9 sensors-22-04018-f009:**
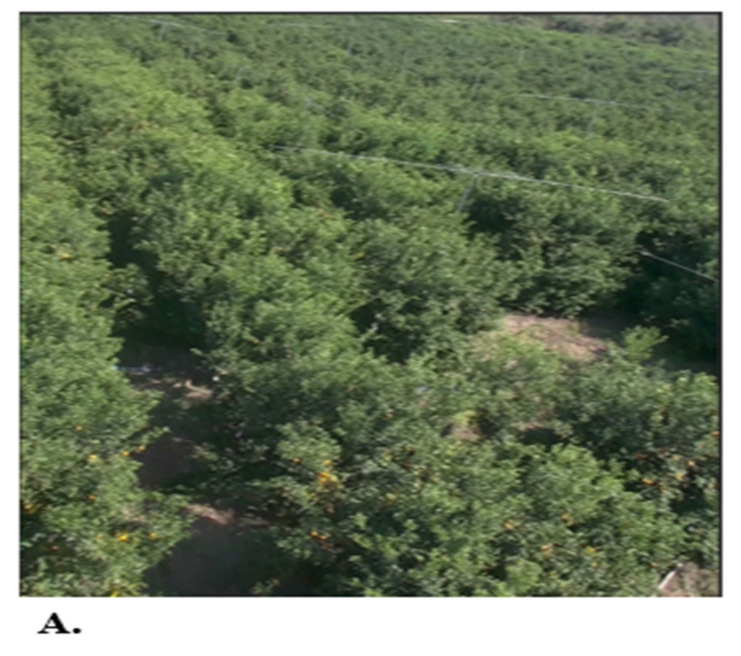
(**A**) digital imagery of the mandarin orchard. (**B**) Isothermal panel showing histogram of pixels and their corresponding CWSI values from the WWARIC.

**Figure 10 sensors-22-04018-f010:**
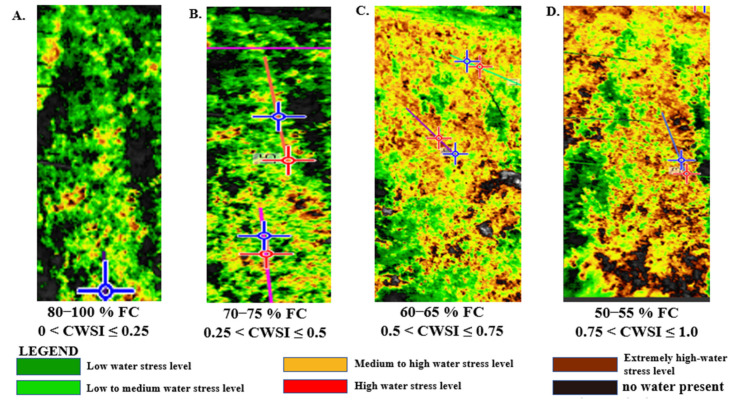
Isothermal imagery of the mandarin orchard depicting various water stress levels and their corresponding range of CWSI values recorded by the WWARIC. (Subset (**A**) is the isothermal image of mandarin under low water stress level (**B**). isothermal image of mandarin subjected to mild water stress (**C**). isothermal image of mandarin under moderate water stress (**D**). isothermal image of mandarin under high water stress.

**Figure 11 sensors-22-04018-f011:**
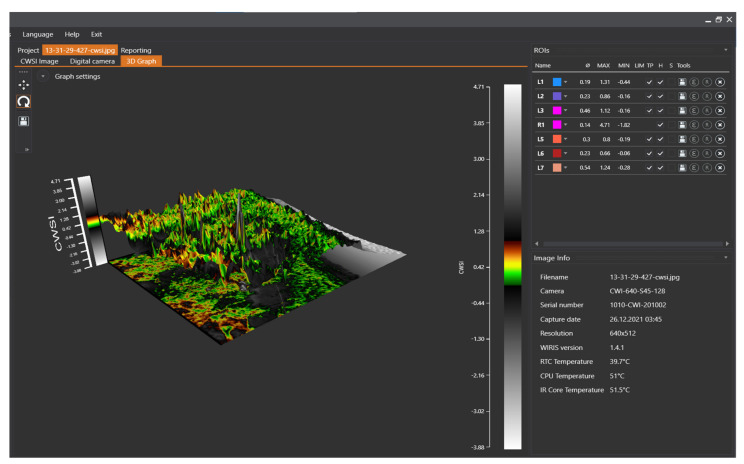
3D graph of the thermal data obtained by the WWARIC.

**Figure 12 sensors-22-04018-f012:**
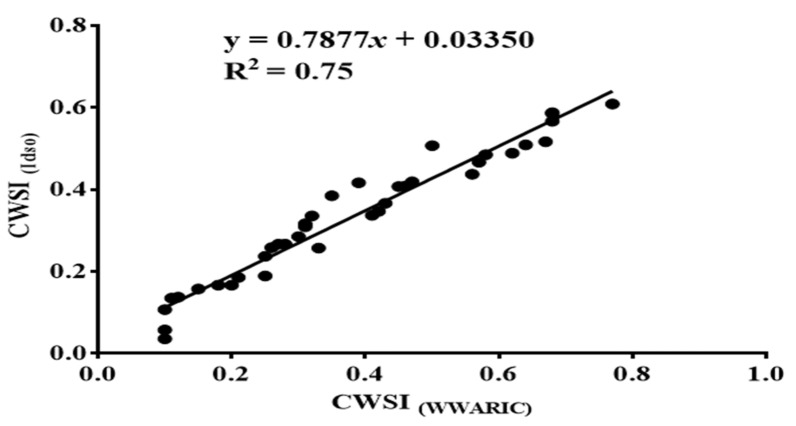
Correlation between CWSI calculated from Idso procedures and CWSI from WWARIC.

**Figure 13 sensors-22-04018-f013:**
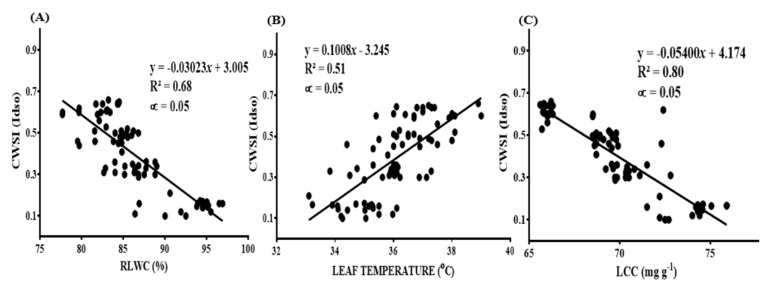
Correlation between (**A**) CWSI_(Idso)_ and RLWC (%) and (**B**) CWSI_(Idso)_ and leaf temperature (°C); (**C**) CWSI_(Idso)_ and LCC (mg g^−1^).

**Figure 14 sensors-22-04018-f014:**
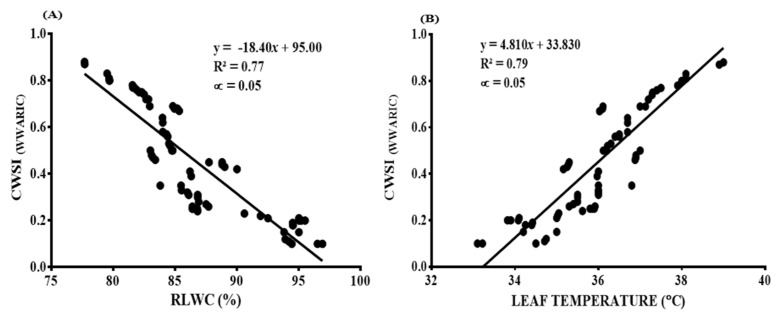
Correlation between (**A**) CWSI_(WWARIC)_ and RLWC (%) and (**B**) CWSI_(WWARIC)_ and leaf temperature (°C).

**Table 1 sensors-22-04018-t001:** General soil physical and chemical properties.

Soil Physical Properties														
Depth (cm)	Textural Analysis									Field Capacity (%)	Permanent Wilting Point (%)		Bulk Density (gcm^−3^)	
	Sand (%)			Clay (%)		Silt (%)		Textural Class						
0–30	72.80			20.00		7.20		Loamy sand		22.60 ± 0.1	10.35 ± 0.1		1.47 ± 0.1	
30–60	73.80			19.20		7.02		Loamy sand		24.20 ± 0.1	10.30 ± 0.1		1.48 ± 0.1	
60–90	69.70			19.10		11.20		Loamy sand		22.10 ± 0.1	10.27 ± 0.1		1.53 ± 0.1	
Average	72.10			19.43		8.47		Loamy sand		22.97 ± 0.1	10.31 ± 0.1		1.49 ± 0.1	
**Soil Chemical Properties**														
**Parameter**		**pH**	**O.M (g/kg)**		**Total N (g/kg)**		**Total K (g/kg)**		**Total P (g/kg)**	**Alkalized N (mg/kg)**		**Available P (mg/kg)**		**Available K (mg/kg)**
Values		4.90	8.83		0.42		2.27		0.37	39.81		85.91		72.89

**Table 2 sensors-22-04018-t002:** Meteorological data during mandarin seasonal development.

Season	Summer	Autumn
Air temperature (℃)	32.5	28.0
Relative humidity (%)	79.3	75.0
Vapor pressure deficit (kPa)	1.5	1.5

Note: (1) Values shown are average seasonal values (22 June 2021 to 30 November 2021). (2) Summer (22 June 2021 to 22 September 2021); autumn (23 September 2021 to 20 December 2021).

**Table 3 sensors-22-04018-t003:** Irrigation water supplied for mandarin in different seasons in 2021.

Season	Irrigation Period (Days)	Irrigation Amount (mm)			
		80–**100**% FC	70–**75**% FC	60–**65**% FC	50–**55%** FC
Summer	June (9 days)	-	-	-	-
	July (31 days)	24.05	17.44	15.03	12.63
	August (31 days)	57.20	41.50	35.70	30.00
	September (22 days)	24.39	18.60	15.60	13.70
	Total for summer	105.64	77.54	66.33	56.33
Autumn	September (7 days)	9.40	6.00	5.80	5.94
	October (31 days)	32.80	24.04	21.20	17.96
	November (30 days)	36.80	26.44	23.07	19.91
	Total for autumn	79.00	56.48	50.07	43.81
Total irrigation amount		184.64	134.02	116.40	100.14

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
