# Peer review of "Real-Time Assessment of Mandarin Crop Water Stress Index"

_sensors, 2022, doi:10.3390/s22114018_

Round 1
Reviewer 1 Report
The paper is quite clumsily written, and the content itself is inadequately presented. Comments can be made on several levels, on methodology (the research methodology is not concisely explained at the beginning of the paper and by the end of the reading more questions are raised than the conclusions of the paper can provide), on water stress (where the data on the analyzed plants are, wouldn't measuring the water content in soil concerning the crop give results cheaper? The title suggests real-time measurement, wouldn't it be interesting to analyze hourly versus seasonal data?) and thermography (why an expensive drone camera is used stationary and why the data is not compared to another manufacturer’s camera; where the data of thermographic analysis are in temperature form). Given the number of authors, I do not believe that all of them read the final version carefully. It follows the CWSI, which is extremely important, but readers were not informed in the abstract that it is the Crop Water Stress Index. Also, the introduced abbreviation WWARI "WORKSWELL WIRIS AGRO R INFRA -25 RED" is described in a way that made me think you were selling Workswell cameras. Speaking of which, there is no information about the specifications of the camera and the price, but the price without the academic discount. Table 1 is missing, the text in the paper is not consistent, some parts are bolded in larger font, images of disturbing proportions are scaled in the form of bitmaps.
Table 2 contains a worrying fact and that is the air temperature of 32.51 °C, how is it possible, with what equipment did you measure the air temperature?
Table 3 the number of decimal places is not coexistent, somewhere one and for most numbers two decimal places. The name of the vector product is used in the places of the scaled product.
I think that the presentation of the equipment and the measurement result of the camera should be done earlier in the text. The rotation of the thermogram in Figure 8 b is confusing, and the data in the figure is not readable. The '' 3D graph of the WWARI camera '' is a bit confusing because I think the software provides 3D, correct me if I am wrong? Figure 9 is interesting, but why is it not compared to the thermograms of the other thermal imaging camera, i.e. actual temperatures, when the research process could be compared to measurements of others. The most important question is why a relatively expensive drone camera stationary and not classic cheap IR detectors are used to monitor withers. Another question is how the authors envisioned the verification and repeatability of the measurements, whether we should buy specialized cameras that we should trust, or whether we can use commonly available, cheaper infrared thermal imaging cameras intended for stationary use. The part I also miss is the description of the plant and its water needs.
Reviewer 2 Report
Review for “Real-time assessment of Mandarin crop water stress index 2”.
General comments:
This research aimed at determining the viability, efficiency and swiftness employing the commercial “WORKSWELL WIRIS AGRO R INFRARED” (WWARI) camera in monitoring water stress and schedule appropriate irrigation regimes in mandarin. Firstly, I agree with the authors about the importance of this topic. In fact, some predictors are most applied throughout the years, due to simple formulation, and characteristics widely common to most launched data collectors. Most of them are considered a tool for researchers and decision-makers. The application in these cases is much more based on default and established preconditions than local or regional tests. Nowadays, with more advanced technologies, researchers and decision-makers can explore more relationships to improve predictions. Experimental design and discussions are adequate. Please, expand the description of the method (how is defined and computed). The presentation of results can be improved. Please, improve the quality of the figures; some are stretched.
For me, it is a good initiative and is on the way to being published. Therefore, I would like to present some adjustments that can help to improve the study and make it published.
Minor comments:
Please, insert a figure showing the study area and the experimental design described.
Table 1 is misaligned and needs corrections.
Table 2: Metrological or meteorological?
Lines 95-97: Please, verify words in bold.
Lines 151-152: Please, correct the size of the text.
Lines 163-202: This section is concentrated within a single paragraph. Please, stratify.
Lines 244-245: Please, correct misalignment.
Line 320: Possible mistake in “sin”. Please, verify.
Line 366: Please, correct the size of the text.
There is a lack of discussion with the state-of-the-art to explain better why the proposed model performed better than others. The Authors may use literature to discuss important topics, highlighting your inferences and valuing results. Also, it is important to discuss if the adopted approach could be applied to other study areas – especially large study areas or areas with a few meteorological stations.
Conclusion: You should explain better why the proposed model performed better than others before highlighting it in your Conclusions section.
Round 2
Reviewer 1 Report
The 'Table 2. Meteorological data during mandarin seasonal development.' has worrying data that needs to be explained or corrected:
Air Temperature (℃)
32.51
28.00
Relative humidity (%)
79.30
75.00
I have not yet encountered measuring instruments that give data to two decimal places, and the authors have not presented an enviable knowledge of the measurement technique.
A thermographic camera can give three decimal places in raw data, but no one would dare write more than one.
